# Immunogenicity of Foot-and-Mouth Disease Virus Dendrimer Peptides: Need for a T-Cell Epitope and Ability to Elicit Heterotypic Responses

**DOI:** 10.3390/molecules26164714

**Published:** 2021-08-04

**Authors:** Rodrigo Cañas-Arranz, Patricia de León, Sira Defaus, Elisa Torres, Mar Forner, María J. Bustos, David Andreu, Esther Blanco, Francisco Sobrino

**Affiliations:** 1Centro de Biología Molecular “Severo Ochoa” (CSIC-UAM), 28049 Madrid, Spain; rcannas@cbm.csic.es (R.C.-A.); pdeleon@cbm.csic.es (P.d.L.); mjbustos@cbm.csic.es (M.J.B.); 2Departament de Ciències Experimentals i de la Salut (DCEXS-UPF), 08003 Barcelona, Spain; sira.defaus@upf.edu (S.D.); fornermar@gmail.com (M.F.); david.andreu@upf.edu (D.A.); 3Centro de Investigación en Sanidad Animal (CISA-INIA), 28130 Valdeolmos, Spain; elisa.torres@inia.es (E.T.); blanco@inia.es (E.B.)

**Keywords:** FMDV, peptide vaccine, dendrimer

## Abstract

An approach based on a dendrimer display of B- and T-cell epitopes relevant for antibody induction has been shown to be effective as a foot-and-mouth disease (FMD) vaccine. B_2_T dendrimers combining two copies of the major FMD virus (FMDV) type O B-cell epitope (capsid proteinVP1 (140–158)) covalently linked to a heterotypic T-cell epitope from non-structural protein 3A (21–35), henceforth B_2_T-3A, has previously been shown to elicit high neutralizing antibody (nAb) titers and IFN-γ-producing cells in both mice and pigs. Here, we provide evidence that the B- and T-cell epitopes need to be tethered to a single molecular platform for successful T-cell help, leading to efficient nAb induction in mice. In addition, mice immunized with a non-covalent mixture of B_2_T-3A dendrimers containing the B-cell epitopes of FMDV types O and C induced similarly high nAb levels against both serotypes, opening the way for a multivalent vaccine platform against a variety of serologically different FMDVs. These findings are relevant for the design of vaccine strategies based on B- and T-cell epitope combinations.

## 1. Introduction

Foot-and-mouth disease (FMD) is a highly transmissible and economically devastating animal disease [1,2,3] for which vaccination and strict regulations on animal movements and markets are the only means for control [4]. The current OIE-approved vaccines consist of the chemically inactivated whole virus emulsified with different adjuvants (reviewed in [5]). In an outbreak in regions where vaccination is not implemented, massive culling of susceptible animals has been the main measure to control the spread of the disease, leading to large economic losses as well as ethical controversy [6]. Conventional polyvalent FMDV vaccines, containing two or more inactivated viruses from different serotypes, aim to match the potentially circulating viruses and the epidemiological status of each region/country. These vaccines have demonstrated their success in eliciting protective immunity against the disease in endemic countries. Nevertheless, inactivated vaccines continue to pose enough disadvantages to have prompted the adoption of non-vaccination (stamping out) policies in the EU and Western countries (reviewed in [7]), making the development of alternative, safer and effective vaccines an important issue.

Subunit vaccines based on well-characterized immunogenic determinants (not the entire pathogen) have emerged as an attractive alternative to conventional formulations. In particular, this strategy includes fully synthetic peptide vaccines that mimic well-defined B- and T-cell epitopes from the infectious agent and induce protection against it [8]. Linear peptides have been associated with low immunogenicity and partial protection [9]. Nevertheless, there are several strategies to address such issues. We have made significant progress towards dendrimeric peptide-based FMD vaccine candidates based on a lysine core that comprises both a continuous B-cell epitope capable of inducing nAb (corresponding to the G-H loop in FMDV VP1 capsid protein), and T-cell epitopes widely evoking CD4+ responses, thus presenting solid and protective responses in natural hosts (reviewed in [10]). These dendrimer constructs, particularly those encompassing two copies of the B-cell epitope and one of the T-cell epitope (B_2_T), can afford levels of protection against FMDV challenge when administered to pigs similar to those induced by conventional vaccines [8]. The initial results showed that the inclusion and specific orientation of the T-cell epitope (i.e., residues 21–35 of FMDV 3A protein) modulates the immunogenicity of the dendrimer. The T-3A epitope was selected from PBMCs of outbred pigs sequentially infected with two FMDVs of different serotypes in order to identify “promiscuous” FMDV T-cell epitopes, and was shown to be capable of stimulating in vitro lymphoproliferation [11]. These responses were abolished by monoclonal antibodies against both class I and class II, with inhibition being higher for class II. The ability of T-3A to stimulate T-cells was confirmed by its ability to specifically induce nAbs in PBMCs from pigs cultured in vitro. Regarding the functional characterization of T-3A, besides its ability to elicit and stimulate in vitro IFNγ-producing cells (demonstrated by ELISPOT), we have additional evidence from intracellular cytokine staining (ICS) of its ability to induce antigen-specific T-cells (memory T-helper cells) (see below). Furthermore, immunization of pigs with B_2,_ i.e., a B_2_T-like construct lacking the T-3A epitope, failed to induce nAbs despite the production of FMDV-specific antibodies. Additional data providing further support for T-3A as a T-cell epitope are beyond the scope of this manuscript. T-3A has been successfully incorporated as a specific FMDV T-cell epitope in other subunit vaccine prototypes [12,13].

In general, it has been shown that juxtaposition of B- and T-cell epitopes within a single molecule is necessary for effective T-cell help (mediated by cognate interactions), leading to efficient induction of specific antibodies against B-cell epitopes [14,15,16] Thus, elucidation of this immunogenic requirement for dendrimer peptides is relevant for the design of future vaccine candidates based on the combination of different B- and T-cell epitopes.

## 2. Materials and Methods

### 2.1. Peptides

The preparation of B-cell epitopes from FMDV O-UKG 11/01, VP1 (residues 140–158), T-cell epitope 3A (residues 21–35), and B_2_ peptide and B_2_T-dendrimers B_2_T-3A from Type C (B_2_T-3A-C) and B_2_T-3A from Type O (B_2_T-3A-O), named B_2_T-3A, has been described previously [8,17,18].

### 2.2. Viruses

The FMDV stock O/UK/11/01 (The Pirbright Institute, UK) was amplified in IBRS-2 cells and the Type C CS8-c1 virus [19] was amplified in BHK-21 cells.

### 2.3. Animals

Groups of 8 or 10 5-to-6-week-old outbred female mice (Swiss ICR-CD1, Envigo) were maintained under standard housing conditions at the CBMSO animal facility. Mice were immunized subcutaneously, twice, on days 0 and 20 or 21 with 100 µg of each peptide emulsified in Montanide ISA 50V2 (Seppic-France) and euthanized on day 40 or 42. Blood samples were collected on days 0, 20/21 and 40/42 post-immunization (pi).

### 2.4. Virus Neutralization Test (VNT)

Neutralization assays were performed as previously described [8]. Briefly, serial 2-fold dilutions of each serum sample were incubated with 100 infection units—50% tissue culture infective doses (TCID_50_)—of FMDV O-UKG 11/01 or CS8-c1 virus for 1 h at 37 °C. Next, a cell suspension of IBRS-2 cells in DMEM was added and the plates were incubated for 72 h. End-point titers were calculated as the reciprocal of the final serum dilution that neutralized 100 TCID_50_ of homologous FMDV in 50% of the wells.

### 2.5. Detection of Anti-FMDV Antibodies by ELISA

Specific antibodies were assayed by ELISA as described [8] using plates coated with peptide B (1 µg) that were incubated with 3-fold dilutions of serum and detected using HRP-conjugated protein A. Plates were read at 450 nm and titers expressed as the reciprocal of the last serum dilution, given an absorbance range of 2 standard deviations above the background (serum on day 0) plus 2 SD.

### 2.6. Statistical Analyses

Differences among peptide-immunized groups in FMDV antibody titers were analyzed using Student’s t-test. Values are cited in the text as means ± SD. All *p* values are two-sided, and *p* values < 0.05 were considered significant. Statistical analyses were conducted using GraphPad Prism Software 5.0.

## 3. Results and Discussion

### 3.1. Intramolecular Combination of B- and T-Cell Epitopes Is Required for Induction of FMDV-Specific Antibodies

We previously showed that immunization of Swiss mice with B_2_T elicits specific IFN- ɣ-secreting T-cells in response to B_2_T and the T-peptide alone but not the B-peptide, supporting the recognition of T3A as a T-cell by these mice [20]. To study whether the T-3A T-cell epitope needs to be included in the same construct with the B-cell peptide to efficiently elicit FMDV antibodies, groups of 10 mice were immunized and boosted (day 20 pi) with 0.1 g/dose of B_2_T-3A, B_2_+T-3A, B_2_ or T-3A. In the group immunized with B_2_T-3A, most mice developed ELISA-detectable Ab titers upon the first injection (1.2 ± 0.6 log_10_), which were boosted by a second dose (9 of 10 mice; 1.5 ± 0.5 log_10,_ Figure 1A,B). Consistent with the lack of T-cell epitopes, the antibody response was significantly lower in animals immunized with B_2_ alone (0.5 ± 0.1 log_10_), with only a single mouse showing titers above 1 upon boosting. No antibody response was detected in animals immunized with T-3A alone. Interestingly, when both epitopes were given separately (B_2_+T-3A), the antibody response was significantly lower than with B_2_T-3A, with ELISA titers above 1 for only two mice on day 42 (0.5 ± 0.6 log_10_) (Figure 1A,B). These results correlated well with neutralizing antibody titers (VNT), which were only consistently found after the second dose in mice immunized with B_2_T-3A (1.7 ± 0.6 log_10_) (Figure 1C).

These results support the need for covalently linking both B- (i.e., B_2_) and T-cell epitopes (i.e., T-3A) within a single molecular entity to provide T-cell help and high titers of anti-FMDV nAbs. This is in tune with previous results showing that T-cell-dependent B-cell activation relies on the inclusion of both epitopes in the same molecular platform, as an efficient antibody response occurred when both epitope types targeted the same APC.

### 3.2. A Mixture of B_2_T-3A Dendrimers Harboring B-Cell Peptides from Different FMDV Serotypes Elicits Similar Titers of Type-Specific Neutralizing Antibodies

Current polyvalent FMD vaccines incorporate inactivated viruses from different FMDV serotypes. To explore the capability of our dendrimer platform to allow heterotypic immunization, we assessed the neutralizing antibodies elicited by a mixture of two B_2_T-3A constructs with different B-cell epitopes. To this end, groups of eight mice were immunized twice with: (i) an equimolar mixture of B_2_T-3A-O + B_2_T-3A-C containing, respectively, the B-cell epitope motifs from Type O FMDV O/UK/11/2001 and Type C CS8-c1 isolate; (ii) B_2_T-3A from Type C (B_2_T-3A-C) or (iii) B_2_T-3A from Type O (B_2_T-3A-O) (Figure 2).

Homologous neutralization assays were used to determine the capacity of sera from Groups ii and iii above to neutralize C-S8c1 or O/UK/11/2001, respectively. In parallel, neutralization by sera from group i (B_2_T-3A mixture) was analyzed. In this experiment, no VNT were observed on day 21 pi (data not shown). When the capacity to neutralize type O FMDV was observed (day 40 pi), similar VNT titers were noticed in mice immunized with B_2_T-3A-O (group iii) or the B_2_T-3A mixture (group i) (2.1 ± 0.8 vs. 2.2 ± 0.8 log_10_) (Figure 2A). Likewise, animals in group ii (given B_2_T-3A-C) produced type C CS8-c1 VNTs similar to those in the B_2_T-3A mixture (group i) (2.2 ± 0.8 vs. 2.4 ± 0.8 log_10_) (Figure 2B). When the neutralizing activity against both viruses was compared, six out of eight animals (75%) showed high and similar titers against both serotypes, although the correlation was not significant (*p* > 0.5) (Figure 2C). Pearson’s correlation coefficient (r = 0.07) between titers to serotypes C and O in each animal suggested a similar response.

Early studies on the immunogenicity of FMDV peptide vaccines showed that linear peptides encompassing B- and/or T-cell epitopes elicited levels of protection that were insufficient for their potential commercial application [9,21,22]. This limitation has been addressed by various improvement strategies, including optimization of the B- and T-cell peptides [13,23,24,25,26] and epitope multimerization [12] achieved through the use of molecular platforms, such as that used in this study, B_2_T, in which B- and T-cell epitopes—the latter was selected as being widely recognized by farm pigs [27]—were jointly and multiply displayed [28]. Indeed, the protection afforded to pigs by B_2_T vaccines has been shown to be superior to that of other molecules also incorporating the B-cell peptide in B_2_T (the G-H loop in VP1) (reviewed in [10]). Here, we have further characterized the properties of B_2_T dendrimer vaccines that are relevant for its use as a field FMD vaccine.

Implementation of efficient vaccination campaigns against FMD requires the use of inactivated viruses that are capable of eliciting protective responses against circulating and emerging FMDVs, which is achieved by including serotype-specific vaccine isolates in vaccine formulations. Thus, because of the wide antigenic range presented by FMDV, an optimal vaccine needs to protect against a wide FMDV spectrum. This is particularly the case for vaccines against type O viruses, which are responsible for major outbreaks in epidemic countries [29]. Initial experiments with lineal peptides indicated that the nAbs elicited were able to neutralize not only the homologous virus, whose sequence was that of the VP1 GH-loop they contained, but also heterologous FMDV isolates [30]. Here, we provide evidence that mice immunized with a non-covalent mixture of B_2_T-3A dendrimers containing the B-cell epitopes of FMDV types O and C induced similarly high nAb levels against both serotypes, opening the way for a multivalent vaccine platform against a variety of serologically different FMDVs. In addition, we showed that B- and T-cell epitopes need to be tethered to a single molecular platform for successful T-cell help, leading to efficient nAb induction in mice.

Overall, these findings are relevant for the design of vaccine strategies based on B- and T-cell epitope combinations against different FMDVs.

## 4. Conclusions

Our data support the need for a single molecular platform to efficiently present the B- and T-cell epitopes included in B_2_T dendrimers, as well as the potential of non-covalent mixtures of B_2_T-3As to induce potent anti-FMDV antibodies against viruses from two different serotypes, which opens the possibility of developing a feasible vaccine platform against a broad spectrum of serologically different FMDVs.

## Figures and Tables

**Figure 1 molecules-26-04714-f001:**
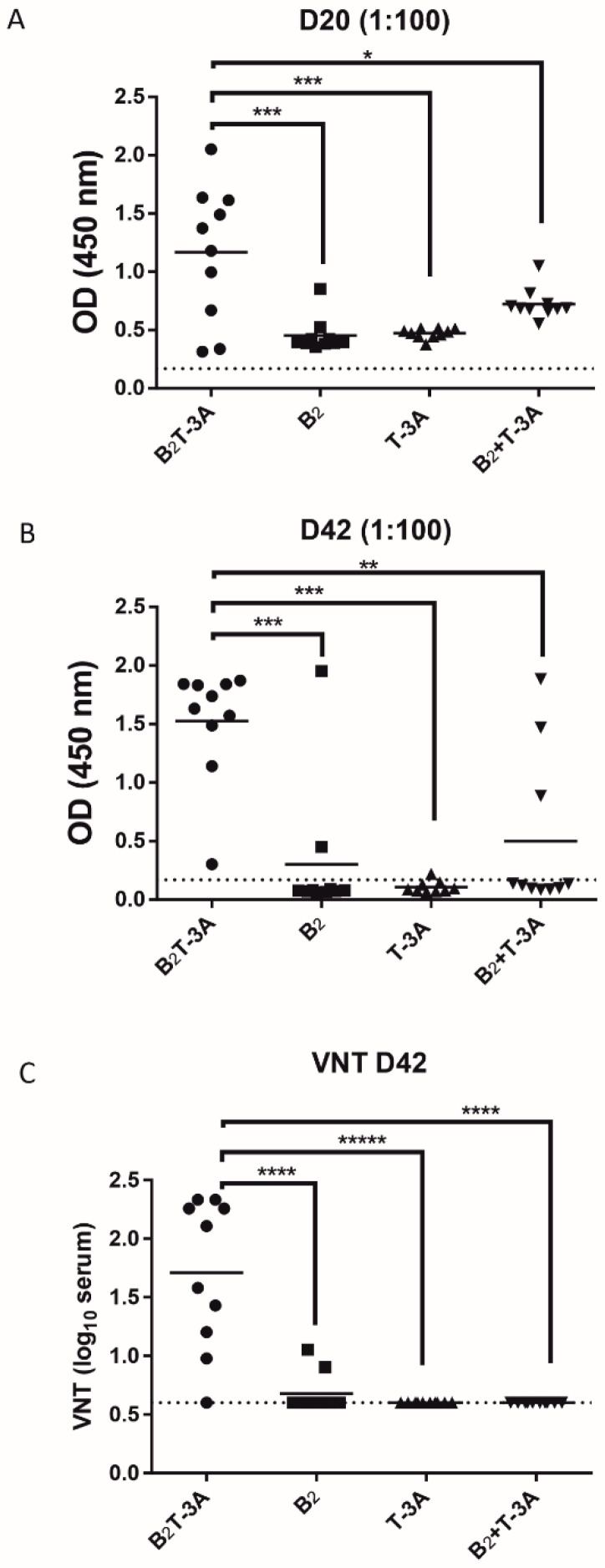
Incorporation of both B- and T-cell epitopes is required for efficient induction of FMDV-specific antibodies by FMDV dendrimers. Antibodies in sera from groups of animals (n = 10 Swiss mice) immunized twice (on days 0 and 20) with 0.1 mg/dose of B_2_T-3A, B_2_+T-3A, B_2_ or T-3A. ELISA titers against B_2_ peptide at 20 days pi (**A**) and 42 days pi (**B**). Virus-neutralizing antibody titers (VNT) on day 42 pi against O/UK/11/2001, a type O FMDV isolate whose sequence is homologous to those of the B_2_ and T-3A peptides (**C**). Each point represents the mean of a triplicate assay for each mouse. Dotted lines indicate the detection limit of the assay. Values are expressed as the reciprocal log_10_ of the serum dilution that neutralized 100 TCID_50_ of FMDV. Statistically significant differences are indicated by asterisks (*) for *p* < 0.05, (**) for *p* < 0.01, (***) for *p* < 0.001, (****) for *p* < 0.0001, and (*****) for *p* < 0.00001.

**Figure 2 molecules-26-04714-f002:**
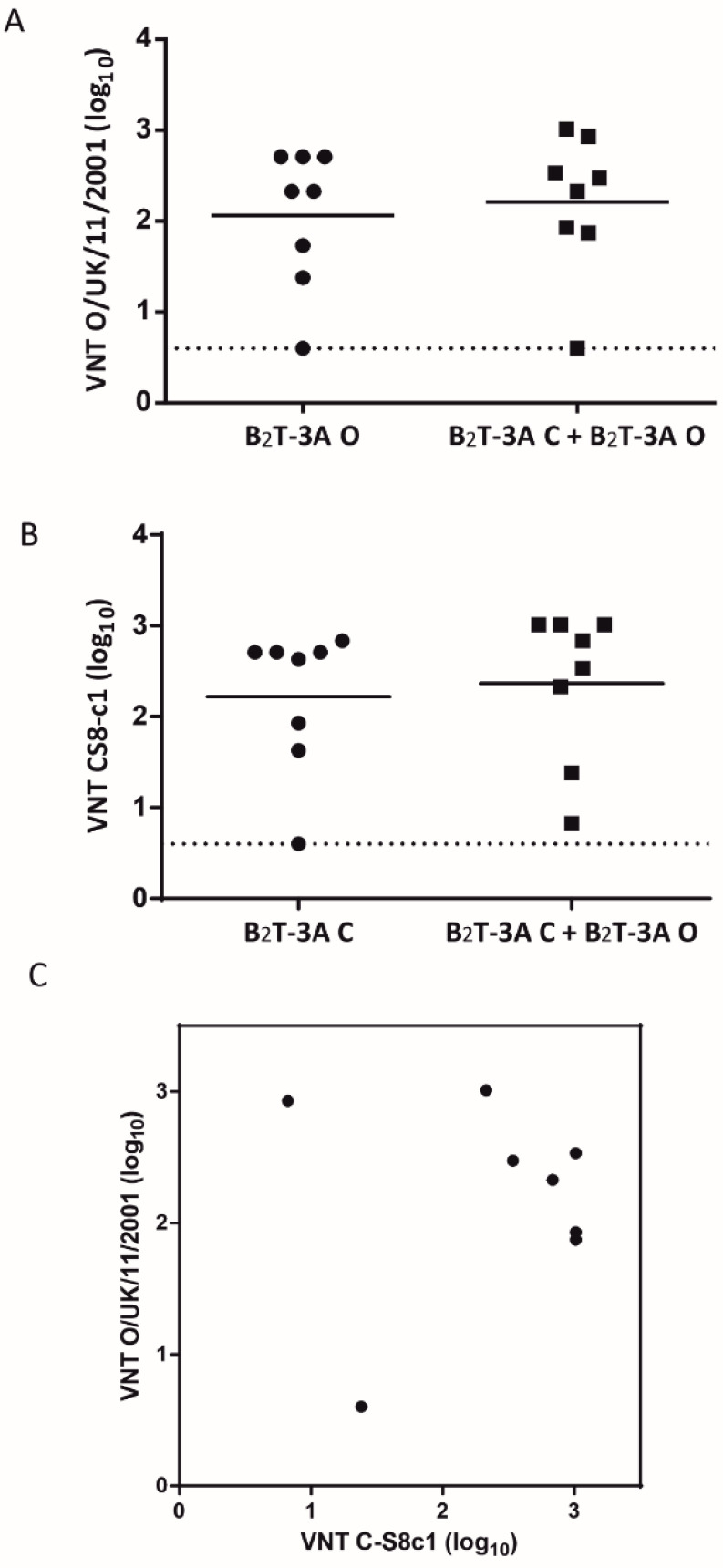
The mixture of two heterotypic B_2_T-3A constructs elicited similar levels of neutralizing antibodies against Type C and Type O FMDVs. Neutralizing antibody titers in sera samples on day 40 pi from mice immunized twice (on days 0 and 21) with an equimolar mixture of B_2_T-3A-O + B_2_T-3A-C or with B_2_T-3A-C or B_2_T-3A-O against Type O FMDV (O/UK/11/2001 isolate) (**A**) and Type C FMDV (C-S8 isolate) (**B**). (**C**) To measure the relationship between antibody titers to serotypes C and O in each animal, Pearson’s correlation coefficient (r) was calculated: r = 0.07. Each point represents the mean of a triplicate assay of each animal (n = 8). Values are expressed as the reciprocal log_10_ of the serum dilution that neutralized 100 TCID_50_ of each FMDV.

## Data Availability

We did not send data to anay database.

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
