# Peer review of "Immunogenicity of Foot-and-Mouth Disease Virus Dendrimer Peptides: Need for a T-Cell Epitope and Ability to Elicit Heterotypic Responses"

_molecules, 2021, doi:10.3390/molecules26164714_

Round 1
Reviewer 1 Report
In this manuscript, Rodrigo Cañas-Arranz et al. demonstrated that the T-3A epitope needs to be included in the same construct with the B-cell peptide for induction of FMDV specific antibodies. In addition, their results revealed that a mixture of B2T-3A-O and B2T-3A-C elicits similar titers of type-specific neutralizing antibodies.
The review is very interesting and lays down useful information. In general, as a brief report, the manuscript was well written.
Minor:
- The author should carefully consider the keywords, mice may not be a good candidate that can be replaced by another word.
- The discussion had better contain more information on the advantages of this study compared to other studies and how is it beneficial to a feasible vaccine platform.
Author Response
1. The author should carefully consider the keywords, mice may not be a good candidate that can be replaced by another word.
Thank you for your suggestion. We have replaced this word by “dendrimer”.
2. The discussion had better contain more information on the advantages of this study compared to other studies and how is it beneficial to a feasible vaccine platform.
To discuss the novelty and advantages of the dendrimeric platform used and the results obtained, the following paragraph has been included in the Results and Discussion section (page 8):
“Early studies on the immunogenicity of FMDV peptide vaccines, showed that linear pep-tides encompassing B- and/or T-cell epitopes elicited levels of protection insufficient for their potential commercial application [9,21,22]. Such limitation has been addressed by various improvement strategies including optimization of the B- and T-cell peptides [13,23-26] and epitope multimerization [12] achieved by the use of molecular platforms, such as that used in this study, B2T, in which B- and T-cell epitopes -the later selected as widely recognized by farm pigs [27]- are jointly and multiply displayed [28]. Indeed, the protection afforded in pigs by B2T vaccines has been shown superior to that of other molecules also incorporating the B-cell peptide in B2T (the G-H loop in VP1) (reviewed in [10]). Here we have further characterized properties of B2T dendrimer vaccines relevant for its use as field FMD vaccines.
Implementation of efficient vaccination campaigns against FMD, requires the use of inactivated viruses capable of eliciting protective responses against circulating and emerging FMDVs, which is achieved by including serotype-specific vaccine isolates into vaccine formulations. Thus, because of the wide antigenic range presented by FMDV, an optimal vaccine needs to protect against a wide FMDV spectrum. This is particularly the case for vaccines against type O viruses, which are responsible of major outbreaks in epidemic countries [29]. Initial experiments with lineal peptides indicated that the nAbs elicited were able to neutralize not only the homologous virus, whose sequence was that of the VP1 GH-loop they contained, but also heterologous FMDV isolates [30]. Here, we provide evidence that mice immunized with a non-covalent mixture of B2T-3A dendrimers containing the B-cell epitopes of FMDVs types O and C induced similarly high nAb levels against both serotypes, opening the way for a multivalent vaccine platform against a variety of serologically different FMDVs. In addition, we show that B- and T-cell epitopes need to be tethered to a single molecular platform for successful T-cell help leading to efficient nAb induction in mice.
Overall, these findings are relevant for the design of vaccine strategies based on B- and T-cell epitope combinations against different FMDVs.”
Reviewer 2 Report
The authors reported a method using a dendrimer display of B- and T-cell epitopes relevant for antibody induction toward FMD vaccines. The authors attempted to use it to facilitate the understanding of immunogenic requirement for dendrimer peptides is relevant for the design of future vaccine candidates based on the combination of different B- and T-cell epitopes. Overall, the idea is interesting; however, the authors need to address the following concerns before considering publication.
1) The structure of the manuscript should check the author guideline with MDPI. The authors fail to provide an informative Conclusion section.
2) In addition, the methods are not adequately described, which made it difficult for other researchers to repeat.
3) The authors should also clarify its novelty by systematically compare their results with their peers. Currently, the discussion lacks of content and looks like a report, which disqualifies a peer-review journal. I hope the authors can improve the manuscript before resubmitting it.
Author Response
1) The structure of the manuscript should check the author guideline with MDPI. The authors fail to provide an informative Conclusion section.
Thank you for your comment. We have included a “Conclusion” section.
2) In addition, the methods are not adequately described, which made it difficult for other researchers to repeat.
Thank you for your comment. We have included a “Materials and Methods” section.
3) The authors should also clarify its novelty by systematically compare their results with their peers. Currently, the discussion lacks of content and looks like a report, which disqualifies a peer-review journal. I hope the authors can improve the manuscript before resubmitting it.
As commented in the responses to Reviewer 1, to discuss the novelty and advantages of the dendrimeric platform used and the results obtained, the following paragraph has been included in the Results and Discussion section (page 8):
“Early studies on the immunogenicity of FMDV peptide vaccines, showed that linear pep-tides encompassing B- and/or T-cell epitopes elicited levels of protection insufficient for their potential commercial application [9,21,22]. Such limitation has been addressed by various improvement strategies including optimization of the B- and T-cell peptides [13,23-26] and epitope multimerization [12] achieved by the use of molecular platforms, such as that used in this study, B2T, in which B- and T-cell epitopes -the later selected as widely recognized by farm pigs [27]- are jointly and multiply displayed [28]. Indeed, the protection afforded in pigs by B2T vaccines has been shown superior to that of other molecules also incorporating the B-cell peptide in B2T (the G-H loop in VP1) (reviewed in [10]). Here we have further characterized properties of B2T dendrimer vaccines relevant for its use as field FMD vaccines.
Implementation of efficient vaccination campaigns against FMD, requires the use of inactivated viruses capable of eliciting protective responses against circulating and emerging FMDVs, which is achieved by including serotype-specific vaccine isolates into vaccine formulations. Thus, because of the wide antigenic range presented by FMDV, an optimal vaccine needs to protect against a wide FMDV spectrum. This is particularly the case for vaccines against type O viruses, which are responsible of major outbreaks in epidemic countries [29]. Initial experiments with lineal peptides indicated that the nAbs elicited were able to neutralize not only the homologous virus, whose sequence was that of the VP1 GH-loop they contained, but also heterologous FMDV isolates [30]. Here, we provide evidence that mice immunized with a non-covalent mixture of B2T-3A dendrimers containing the B-cell epitopes of FMDVs types O and C induced similarly high nAb levels against both serotypes, opening the way for a multivalent vaccine platform against a variety of serologically different FMDVs. In addition, we show that B- and T-cell epitopes need to be tethered to a single molecular platform for successful T-cell help leading to efficient nAb induction in mice.
Overall, these findings are relevant for the design of vaccine strategies based on B- and T-cell epitope combinations against different FMDVs.”
Reviewer 3 Report
The manuscript by Cañas-Arranz et al showcases the ability of dendrimeric B2T-3A peptides constructs to induce a robust antibody response against FMDV of serotypes O and C when provided as single or divalent vaccines. The results also demonstrated the importance of intramolecular presentation of both, T-cell and B-cell epitopes to induce a significant neutralizing antibody response following a prime-boost vaccination regimen.
- The time when a boost vaccination was given to animals should be described in the manuscript and also state in figure legends the fact that animals were vaccinated twice. That will clarify that day 20pi follows a prime vaccination and that 40pi is indeed a time post boost.
- Figure 1 will benefit from inclusion of data on T-cell responses, or if already published give reference to it . There is no evidence presented here that the T-3A molecule is capable of inducing a T cell response, only assumed it does. How you exclude the possibility that the T-3A molecule in the dendrimer is assisting not as an T epitope but perhaps by presenting the B-cell epitope in a more effective conformation? A statement will help interpretation of the results.
- If challenge of vaccinated animals has been conducted on B2T-3A dendrimers described here adding that information in the manuscript will increase its impact. Again elaborate on it in the result section as it is well known that neutralizing antibodies could be induced by a variety of FMDV peptides presenting the G-H loop in VP1, but not all of those responses are strong enough and capable of inducing a protective response following challenge. The readers should understand what makes dendrimers a better option, something that could improve the current version of this manuscript.
Author Response
1. The time when a boost vaccination was given to animals should be described in the manuscript and also state in figure legends the fact that animals were vaccinated twice. That will clarify that day 20pi follows a prime vaccination and that 40pi is indeed a time post boost.
We have included that information in figure legends and also in a “Materials and Methods” section.
2. Figure 1 will benefit from inclusion of data on T-cell responses, or if already published give reference to it . There is no evidence presented here that the T-3A molecule is capable of inducing a T cell response, only assumed it does. How you exclude the possibility that the T-3A molecule in the dendrimer is assisting not as an T epitope but perhaps by presenting the B-cell epitope in a more effective conformation? A statement will help interpretation of the results.
Thanks for this comment. We previously showed that immunization of Swiss mice with B2T elicits specific IFN- ɣ secreting T-cells in response to B2T and the T-peptide alone but not by the B-peptide, supporting the recognition of T3A as a T-cell by these mice (Blanco et al., 2013; Clin Dev Immunol. 2013; 2013: 475960 new reference [20]). This is now commented in the text, page 3, line 114. Unfortunately, and due to logistic limitations we could not perform this analysis in the experiments presented here.
3. If challenge of vaccinated animals has been conducted on B2T-3A dendrimers described here adding that information in the manuscript will increase its impact. Again elaborate on it in the result section as it is well known that neutralizing antibodies could be induced by a variety of FMDV peptides presenting the G-H loop in VP1, but not all of those responses are strong enough and capable of inducing a protective response following challenge. The readers should understand what makes dendrimers a better option, something that could improve the current version of this manuscript.
According to this comment, a comment of the protection observed in pigs immunize with B2T dendrimers in now included in page 8, line 258:
“Indeed, the protection afforded in pigs by B2T vaccines has been shown superior to that of other molecules also incorporating the B-cell peptide in B2T (the G-H loop in VP1) (reviewed in [10]).”
Round 2
Reviewer 2 Report
Accept